



**Assessing the role of planetary and gravity waves on the vertical structure of ozone over**
**central Europe**

4                              Peter Križan
6            *Institute of Atmospheric Physics, Czech Academy of Sciences*
7                          *krizan@ ufa.cas.cz*

**Abstract**
*Planetary and gravity waves play an important role in the dynamics of the atmosphere. They*
*are present in the atmospheric distribution of temperature, wind and ozone content. These*
*waves are detectable also in the vertical profile of ozone and they cause its undulation. One of*
*the structures occurring in the vertical ozone profile is laminae, which are narrow layers of*
*enhanced or depleted ozone concentration in the vertical ozone profile. They are connected*
*with the total amount of ozone in the atmosphere and with the activity of the planetary and the*
*gravity waves. The aim of this paper is quantifying these processes in the central Europe. We*
*compare the occurrence of laminae induced by planetary waves (PL) with the occurrence of*
*these induced by gravity waves (GL). We show that the PL are 3-5 times more frequent than*
*the gravity wave ones.   There is a strong annual variation of PL, while GL exhibit only a very*
*weak variation. With the increasing lamina size the share of GL decreases and the share of*
*PL increases. The vertical profile of lamina occurrence is different for small planetary wave*
*and gravity wave laminae. The trend of large lamina occurrence frequency is given by the*
*trend in PL, not by GL.*
**Key words:** ozone lamina; vertical ozone profile, planetary wave activity, gravity waves
**1. Introduction**
32       There are various structures in the vertical profile of ozone affected by the activity of the
planetary and gravity waves. Ones of them are narrow layers of the enhanced or depleted
ozone concentration in the ozone vertical profile, which are called ozone laminae. The first
investigation of these structures was made by Dobson (1973), who found that they occur
predominantly in a cold half of the year. Similar findings were obtained also by Reid and
Vaughan (1991). They observed the maximum occurrence of laminae in the lower
stratosphere at heights around 14 km.  Above the ozone profile maximum their occurrence is
rare.
40       The existence of laminae was confirmed by lidar and satellite measurements (Bird et al.,
1997, Orsolini et al., 1997, Kar et al., 2002).  They were found also in water vapour in the
stratosphere (Teitelbaum et al., 2000). The dynamics of the stratosphere plays a crucial role in
a lamina formation. This finding was confirmed by the ability of dynamical models to capture
these narrow layers (Manney et al., 2000, Orsolini et al., 2001). The number of large laminae
is strongly correlated with the total ozone content and it is the reason why we have been
interested in laminae (Krizan and Lastovicka, 2005).
47       The laminae are not only the indicator of the atmospheric ozone content but also they are
connected with the gravity and planetary wave activity. Teitelbaum et al. (1995) developed a
identification procedure which enable us to detect the planetary and gravity wave activity in
the ozone vertical profile. In this paper we apply this method to ozone laminae and each



lamina we sort to the one of the following groups: laminae induced by gravity wave activity
(GL), by planetary wave activity (PL) and laminae which are neither induced by the gravity
waves nor by the planetary waves. Similar method was used by Grant et al., (1998) and Pierce
and Grant (1998) but only for the Wallops Island station. The aim of this paper is finding the
characteristics of GL and PL in central Europe in the period 1970-2016.  At first we test if the
Teitelbaum method is suitable for central Europe. Next the annual variation of GL and PL is
examined. Then we explore the dependence of lamina composition on their size. We also
compare the vertical distribution of GL and PL. We deal with their trends. The content of this
paper is as follows:  section 2 describes methods and data, section 3 gives results, in section 4
the results are discussed and the last section is conclusions.
**2. Methods and data**
Now we shortly describe the lamina searching procedure. Each positive lamina consists of
the three main points: the lower minimum, the main maximum and the upper minimum. The
depth of lamina must be between 500 and 3500 m due to the vertical resolution of the
ozonosondes (lower limit) and due to the fact that the ozone lamina is a narrow layer of the
enhanced ozone concentration (upper limit). The size of laminae is given as a difference
between the ozone concentration in the main maximum and the average concentration from
both minima. More about the lamina searching procedure can be found in (Krizan and
Lastovicka, 2004) and (Lastovicka and Krizan, 2005).
The method used in this paper for the searching the activity of gravity and planetary waves
in the ozone profile is a modification of the methods given by Teitelbaum et al. (1995). Figure
1 (upper panel) shows the real ozone profile at Hohenpeissenberg from February 2, 1970. We
use the linear interpolation with the step 50 m for the approximating the ozone profile with
the high vertical resolution. Then the 50 point moving average (2500 m in vertical) is applied
to this real profile to obtain the smooth profile. This smooth profile is also displayed in fig.1
(upper panel). The same procedure is applied to the potential temperature and the results are
given in fig. 1 (lower panel). In the next step we compute the differences between the high
resolution profile and the smooth profile for the ozone partial pressure (fig 2 upper panel) and
the potential temperature (fig 2 lower panel). The differences are much higher for the ozone
profile than for the potential temperature profile. The differences in the vertical gradients of
the ozone partial pressure and the potential temperature must be taken into account. So we
must apply the following correction factor to the potential temperature perturbations:
$R\,(z) = [(1/O_{3avg})*(dO_3/dz)]*[(1/\Theta_{avg})*(d\Theta/dz)]$                    (1, 1)
where $O_{3avg}$ ($\Theta_{avg}$) is the average ozone partial pressure (potential temperature) profile in the
layer with the width dz. The vertical distribution of this correction is given in fig.3 (upper
panel). The correction is the highest in the lower stratosphere where the vertical gradient of
ozone is strong. Above 20 km we observe the negative values of this factor, which is
predominantly given by the negative gradient of the ozone partial pressure and the strong
positive gradient of the potential temperature. When we multiply the potential temperature
perturbations with this correction, we obtain the perturbations, which are shown in fig. 3
(lower panel). These new perturbations are not similar to that given in fig.2 –lower panel. In
each point of the high resolution ozone profile we compute the correlation coefficient between
the ozone perturbations and the scaled potential temperature perturbation up to 5 km above
this point.  The vertical dependence of this correlation coefficient from the ground to the point





which is situated 5km below the highest ozone profile point is seen in fig.4. If the correlation
coefficient is greater than 0.7, the vertical ozone profile in this point is influenced by the
gravity waves. In fig 4 the correlations are higher than 0.7 at some altitudes above 5 km and
below 15 km.   If the lamina maximum is situated in this high correlation area we conclude
this lamina is induced by the gravity waves. On the other hand,  if these correlations are low
(between -0.3 and 0.3), we consider the ozone profile to be  influenced by the planetary waves
in this point (from 17 to 22 km on fig. 4) and again if there is a lamina maximum there we
consider this lamina as the one induced by the planetary waves.  When the correlation
coefficient is above 0.3 and below 0.7 or below -0.3 we are not able to evaluate what type of
laminae is present and call them indistinguishable laminae. The boundary values of
correlation coefficients were taken from Teitelbaum et al. (1995)
We apply this procedure to the following European midlatitudes stations:
Hohenpeissenberg (Germany, 1970-2016, 5166 files), Payerne (Switzerland, 1970-2016, 5998
files), Uccle (Belgium, 1970-2015, 6221 files), Lindenberg (Germany, 1975-2013, 2380 files)
and Legionowo (Poland, 1979-2016, 1728 files). These data were taken from WOUDC
Toronto (http://woudc.org/archive/Archive-NewFormat/).
**3. Results**
**3.1. Performance of method**
At first we must answer the question if the procedure used in the paper is successful in
partitioning of laminae to the groups.  If the procedure is suitable, the number of the
indistinguishable laminae cannot be very high. The performance of this procedure is given in
tab.1 for Hohenpeissenberg for each month and for all laminae regardless the size. The results
at the other stations are very similar. From this table we see that approximately 47 % of all
laminae are PL, while GL laminae formed about 10 % and the share of indistinguishable
laminae is about 43 %. It means more than 50 % of all laminae can be divided into the
laminae induced by the gravity or the planetary wave activity. So we can conclude this
procedure is successful in lamina partitioning, because nobody can expect only GL and PL
will be present and no indistinguishable laminae. Practically there is no yearly course in the
lamina composition.
**3.2. Annual variation of laminae induced by the gravity and the planetary wave activity**
Figure 5 shows the annual variation of the number of laminae larger than 2 mPa for
GL and PL at all stations used in this paper. The group of lines with the strong annual
variation with maximum in winter and minimum in summer/autumn are PL while the lines
with the only very weak variation belong to GL. This different behaviour of the annual
variation is the evidence that the both type of laminae are formed by different processes.
**3.3. Dependence of lamina type on the size of laminae**
In this section we deal with the lamina type occurrence frequency in the selected
classes of lamina size. The laminae were sorted to the following groups: small (<1 mPa),
medium size (1-4 mPa) and large (>4 mPa) and in each group we found the occurrence





frequency of different types of laminae. The results are presented in fig.6. The results are
almost identical for all stations. The share of GL is decreasing with the increasing size and the
opposite is true for PL. The performance of used procedure increases with the increasing
lamina size (the share of indistinguishable laminae decreases). The gravity waves are able to
produce predominantly small laminae, while the planetary waves produce also the large ones.
Similar results were also obtained by Teitelbaum et al. (1995).

**3.4. Vertical dependence of the occurrence of advection and gravity wave laminae**

160       Now we examine the altitudinal dependence of occurrence of GL and PL at the
stations used in this paper for all seasons. March, April and May form spring, June, July,
August are summer months, September, October and November are the autumn ones and
December, January and February is winter.   We divided the ozone vertical profile into 2 km
wide intervals and in each interval we search for the lamina occurrence. The results are
displayed as the percentage of all laminae which occur in the individual altitude interval.  We
grouped laminae into two groups: small (<2 mPa) and large (>2 mPa) and in each group we
are searching for the lamina occurrence. The results are displayed only for the station
Hohenpeissenberg, because at the other stations the results are similar. The winter results are
given in fig. 7 for the large (upper panel) and the small (lower panel) laminae. The large
laminae have similar behaviour both for GL and PL.  Their maximal occurrence is observed
in the lower stratosphere and there are no large laminae in the troposphere. On the other hand
the occurrence of the small laminae is different. GL have maximal occurrence in the
troposphere where the occurrence of PL is small.  Small PL have the maximal occurrence in
the lower stratosphere, where the small gravity wave laminae are rare. In the troposphere
there is local minimum in small PL and the main maximum in the small gravity wave
occurrence. Spring (fig.8) behaviour of the lamina occurrence is similar to the winter one. In
summer (fig.9) the large GL have broad stratospheric maximum and the smaller maximum is
observed in the troposphere. Large PL have sharper stratospheric maximum and they are very
little present in the troposphere. Small PL maximum is again observed in the lower
stratosphere.  Small GL have bimodal vertical distribution with one maximum in the lower
stratosphere (similar to the advection one) and the other peak is observed in the troposphere.
At the nearly same height we observe local minimum in small PL and maximum in the
gravity wave ones. In autumn (fig.10) the behaviour of large laminae is a bit similar to the
summer one and the main maximum in occurrence of small GL is higher than that of the
laminae induced by the planetary wave.

**3.5 Trend of the large laminae**

190       Now the long-term of the large laminae occurrence (larger than 4 mPa) is investigated.
The results are shown in fig. 11. A change in the trend of the PL in the mid-1990s is seen.
Before the mid-1990s the negative trend is observed, while after this point the positive one is
present. This fact confirms the findings of Krizan and Lastovicka (2006). But this is not the
main massage of this paper. The main massage of this paper concerning the trend is the
following: we observe a huge difference in the long- term trend between  GL and PL:  trend
of PL has the  sharp change in the mid-1990s,  while the  GL has small significant negative
trend in the period 1970-2016 with no trend change in the mid-1990s.



## 4. Discussion

We found the occurrence frequency of PL to be about 4-6 times larger than that of GL The most frequent way of formation of the laminae induced by planetary waves is vertically different advection of air with the various ozone content (Manney et al., 2000, Tomikawa et al., 2002). In this process we observe transformation of the horizontal gradient of the ozone concentration into the vertical one. The air with the high ozone concentration comes to the central Europe in winter from the edge of the polar vortex (Orsolini et al., 2001). On the other hand the low ozone air has its origin inside the polar vortex and it is transported to the mid latitudes (Reid and Vaughan, 1991) or it is the air from the low latitudes where ozone concentration is low (Orsolini et al., 1995).

The strong source of gravity waves is orography (Smith et al., 2008), especially passing the air through a mountain range when the gravity waves occur in the downwind side of the ridge. For stations used in this paper the most important mountains are the Alps. All stations are situated to the north from these mountains. Because the prevailing winds are from the west, these stations seldom are situated on the leeward side of the Alps and thus the share of gravity wave laminae are practically the same for all stations. The same is true for the laminae induced by planetary waves. In this case all stations are practically under the same conditions. So we cannot expect large interstation differences in lamina partitioning. It will be reasonable to do this investigation at the stations which lie on the leeward side of mountains or at stations which are in hot spots of the gravity wave activity (Sacha et al., 2016). The other sources of the gravity waves are jet stream and convection (Guest et al. 2000; Yoshiki et al. 2004). Their conditions are the same for all stations used in this study. In the troposphere the stratosphere-troposphere exchange may cause the positive laminae and in the stratosphere this exchange may lead to formation of negative laminae (Kritz, 1991).

Laminae greater than 2 mPa occur very predominantly in the stratosphere where the ozone concentration is high. When the ozone concentration is high, the probability of large lamina formation increases. The confirmation of this rule is also the yearly course of PL where the maximal occurrence is observed when the ozone concentration is the highest (winter and spring). On the other hand in the troposphere we observe neither the PL large laminae nor the large GL due to small ozone concentration. Similarly we observe less large PL in the stratosphere in summer and fall. This dependence of the lamina occurrence on the background ozone concentration is valid only for PL, not for the gravity wave ones.

For the laminae smaller than 2 mPa the situation is different. We observe the differences in the vertical distribution of PL and GL. In winter the maximal occurrence is observed in the lower stratosphere in the case of PL, while gravity wave laminae have its occurrence maximum in the tropopause. In spring the small GL maximum lies lower than in winter. In summer the occurrence distribution has bimodal structure with one maximum in the troposphere and the other one in the stratosphere. In fall the stratospheric mode is dominant.

In summer and fall there is no polar vortex. Vortex remnants (Durry et al., 2005) may form the positive laminae in the stratosphere while the advection of air from low latitudes (Koch et al., 2002) creates layers with the low ozone concentration.

In the troposphere the situation is different. Positive laminae are created by various processes: the stratosphere-troposphere exchange (Manney et al., 2000), the advection of polluted air from the boundary layer (Oltmans et al, 2004; Collete et al., 2005) or in situ ozone production (Li et al., 2002). Tropospheric gravity waves occur predominantly in the transition region from the troposphere to the stratosphere where there is a strong change in the atmospheric stability

The vertical resolution of the ozone vertical profiles used in this paper can modify the results. Its value was obtained as an average vertical distance between two adjacent points of

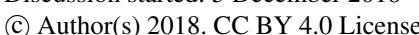  

the ozone profile in a certain year. Its long-term evolution in the period 1970-2016 is given in
fig.12. At the majority of stations the resolution increases (vertical distance decreases) in the
period 1970-2016. And thus we must ask the question if this resolution change has effect on a
number of laminae detected in the profile. We have computed correlation coefficient between
the yearly values of lamina number and vertical resolution. If these correlations are significant
the resolution influences the lamina number and vice versa. We did the correlations for the
following groups of laminae: small (<1 mPa), medium (1-4 mPa) and large (>4 mPa). The
results are shown in tab.2. The number of small laminae is strongly correlated with vertical
resolution. It means the numbers of small laminae are affected by the resolution. With
increasing size of laminae these correlations decrease. For large laminae the results are station
dependant. These results are a bit surprising because one expects negative correlations of
lamina number with resolution and these negative correlations were observed only for small
laminae. For the explanation of these results we must look at the average lamina depth in
small, medium, and large laminae (table 3), which was obtained for the best vertical
resolution (below 100 m). We can see the increase of lamina depth with increasing size. When
the depth of laminae is small (small laminae) the vertical resolution strongly influences the
lamina number, because with decreasing resolution the number of detected laminae decreases.
On the other hand the average depth of large laminae is above the worst vertical resolution
(800 m- fig.12) and so the increasing resolution does not influence significantly the number of
detected laminae.
Our paper is based on the lamina searching procedure introduced by Teitelbaum et al. (1995).
In their paper no climatological results are presented. They illustrated    the method for
partitioning of laminae for several case studies. The goal of our paper is to use this method for
obtaining the climatological results from the mid-Europe ozonosonde stations. Similar
searching method was used by Grant et al. (1998) and Pierce and Grant (1998) but for tropical
and low latitudes stations. The authors found rare occurrence of PL and majority of laminae
was induced by gravity waves. We found more PL compared to the gravity induced ones,
because our investigation was done in middle latitudes, not in the low and tropical ones. The
activity of planetary waves is stronger in mid latitudes compared to the low and equatorial
ones.
**5. Conclusions**
The main results of this paper are:
• The most often the laminae are induced by the planetary wave activity (45-50 %),
following by the indistinguishable ones (about 40 %). The share of the gravity wave
laminae is about 10 %.
• There is a pronounced annual variation in the occurrence frequency of PL, while there
is no such variation for GL
• With increasing lamina size the share of gravity wave and indistinguishable laminae
decreases while the share of the planetary wave laminae increases.
• The vertical distribution of lamina number for large laminae has maximum in the
stratosphere while the distribution of small laminae is type and season dependant.
• There are huge differences in trend patterns of PL and GL in the period 1970-2016.
**Competing interests**
The author declare that he has no conflict of interest



302 .

**Acknowledgement**


Support by the Grant Agency of the Czech Republic via Grant 18-01625S is acknowledged.

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



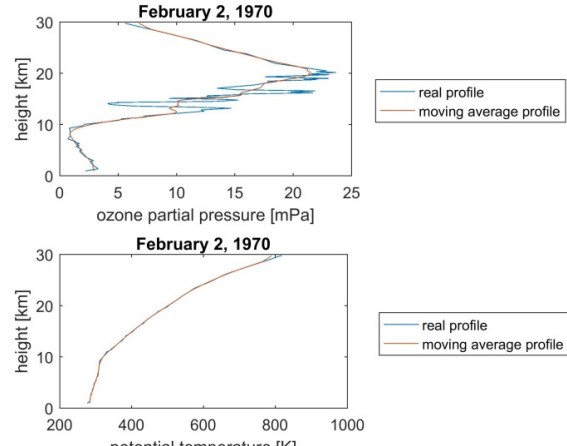

**Figure 1:** Real and smooth ozone (upper panel) and potential temperature (lower panel)
vertical profile at the Hohenpeissenberg from February 2, 1970.


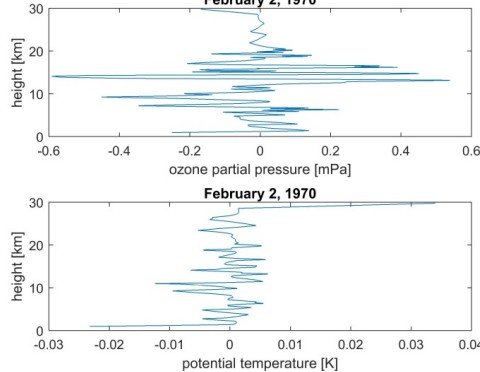

**Figure 2:** Differences between real and smooth vertical profile from February 2 , 1970   for
ozone (upper panel) and potential temperature (lower panel)





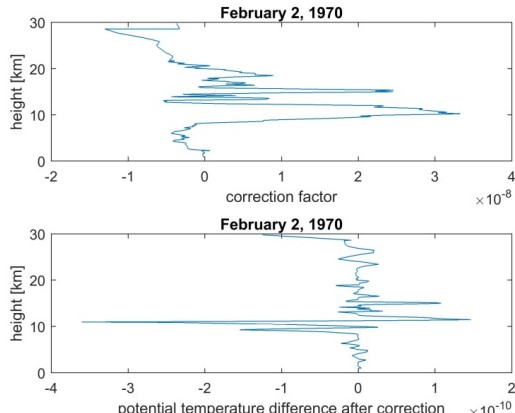

**Figure 3:** Vertical profile of potential temperature correction factor (upper panel) and vertical
profile of differences between real and smooth potential temperature profile (lower panel)
after correction.

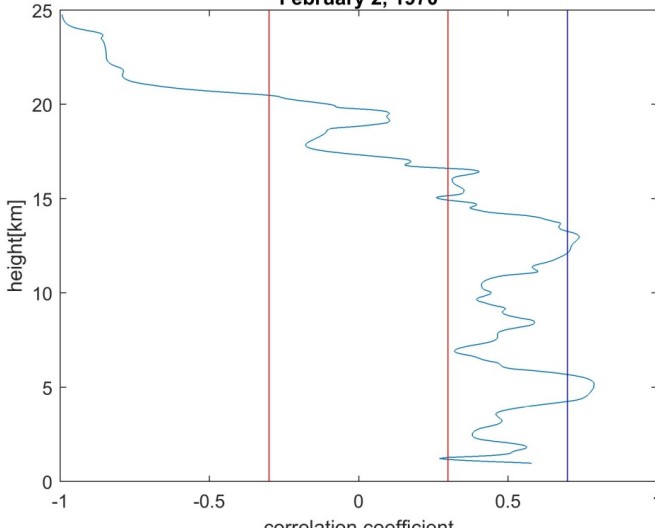

**Figure 4:** The vertical profile of correlations between the corrected potential temperature
differences and the ozone differences from February 2, 1970 at Hohenpeissenberg. The red
vertical lines are the borders for the laminae induced by the planetary waves and the blue
vertical line is the border for gravity wave ones.



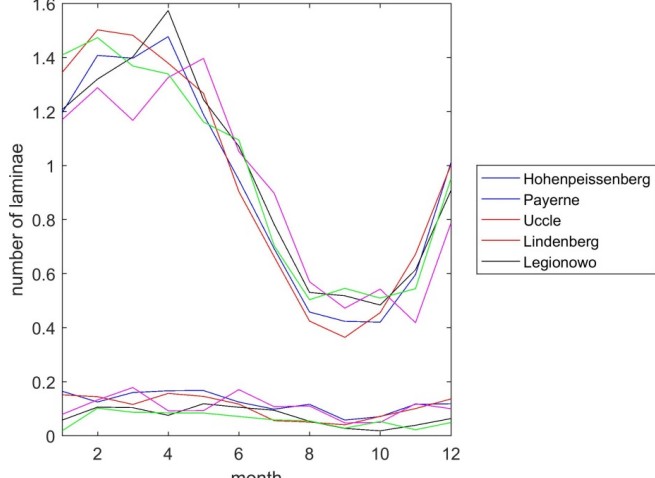

**Figure 5:** The annual variation of the lamina number per ozone profile for PL (group of lines
with the strong variation) and for GL (group of lines with the weak variation) at the European
ozonosonde stations.

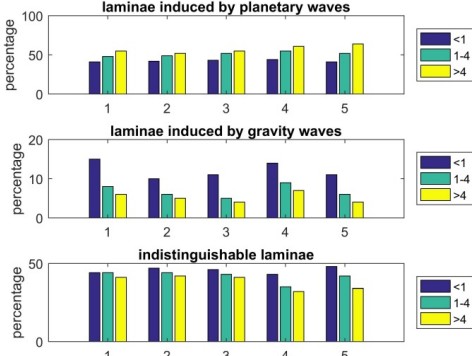

**Figure 6**: The dependence of the lamina composition on a lamina size for PL (upper panel),
GL (middle panel) and indistinguishable laminae (lower panel) at the European stations (1-
Hohenpeissenberg, 2 – Payerne, 3 – Uccle, 4 – Lindenberg, 5 – Legionowo)



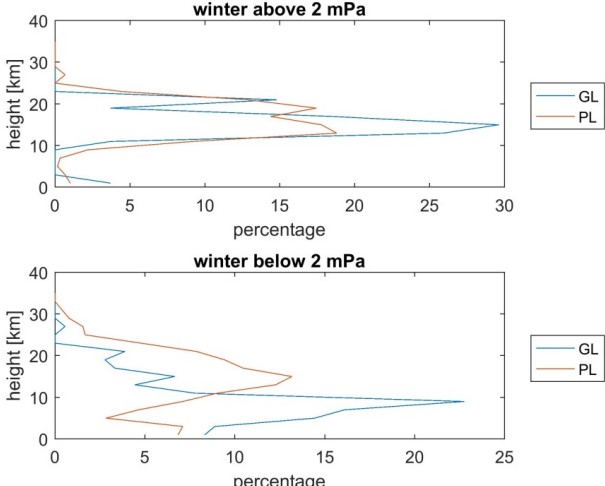

**Figure 7:** The vertical dependence of the occurrence of the laminae induced by the gravity waves and the ones induced by planetary waves at Hohenpeissenberg in the period 1970-2016 in winter in terms of percentage of all GL and all PL.

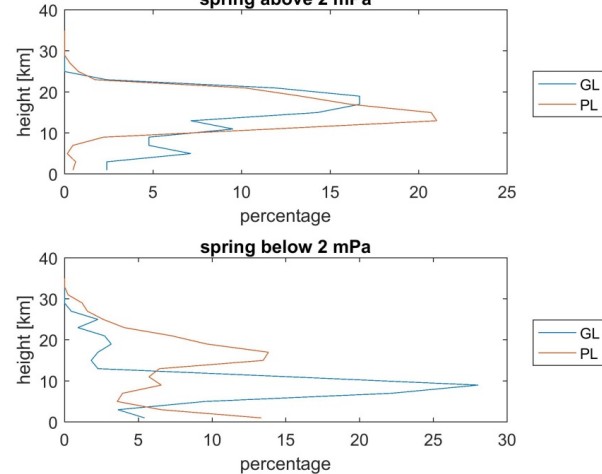

**Figure 8:** The same as fig.7 but for spring





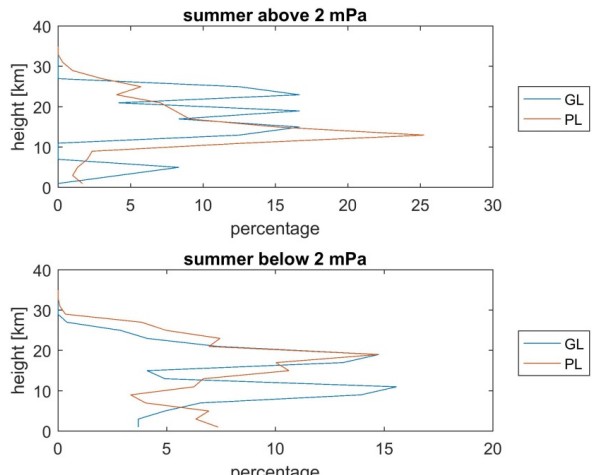

**Figure 9:** Vertical dependence of lamina occurrence in summer.

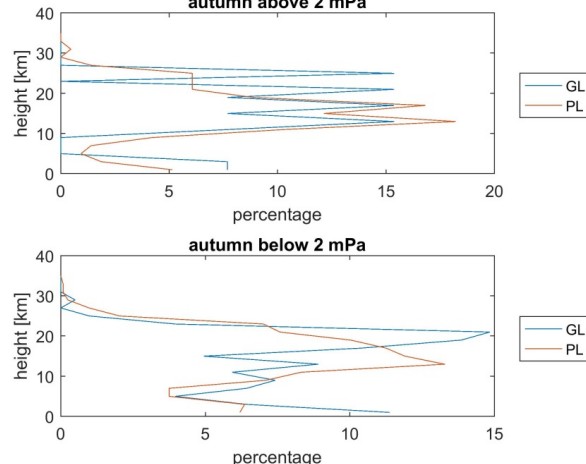

**Figure 10:** The same as fig. 9, but in autumn.



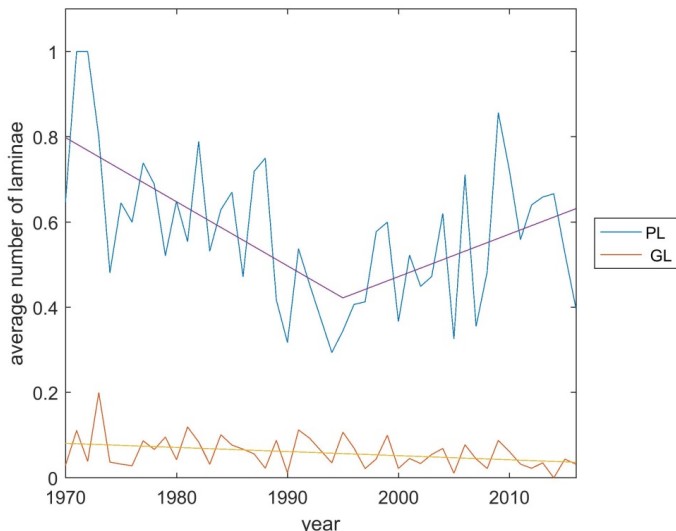

**Figure 11:** Trend of the number of laminae per ozone profile induced by the planetary waves (PL) and by the gravity waves (GL) for laminae greater than 4 mPa at Hoheinpeissenberg in the period 1970-2016.

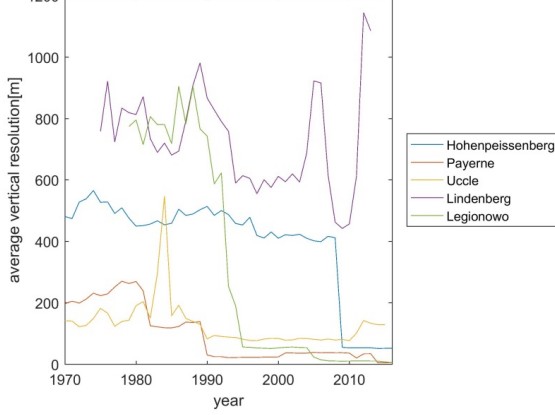

**Figure 12:** Long term evolution of average vertical resolution of profiles at the European ozonesonde stations.



|  | January | February | March | April | May | June | July | August | Sept | Oct | Nov | Dec |
|---|---|---|---|---|---|---|---|---|---|---|---|---|
| advect | 48 | 49 | 48 | 48 | 45 | 41 | 44 | 46 | 47 | 46 | 47 | 48 |
| gravity | 10 | 10 | 11 | 10 | 11 | 11 | 10 | 11 | 10 | 11 | 9 | 10 |
| undist | 42 | 41 | 41 | 42 | 44 | 48 | 46 | 43 | 43 | 43 | 44 | 42 |

**Table 1:** Monthly composition of laminae (%) at Hohenpeissenberg in the period 1970-2016 (advect- advection laminae, gravity – gravity waves laminae, undist- undistinguishable laminae)

|  | <1mPa | 1-4 mPa | >4 mPa |
|---|---|---|---|
| Hohenpeissenberg | **-0.95 /-0.68** | **-0.57/0.55** | -0.09/**0.25** |
| Payerne | **-0.49/-0.37** | **-0.50/0.29** | **0.32/0.58** |
| Uccle | **-0.66/-0.61** | **0.57**/-0.07 | 0.00/0.16 |
| Lindenberg | **-0.79/-0.51** | **-0.88/-0.54** | **-0.76**/0.14 |
| Legionowo | **-0.81/-0.80** | **-0.77**/-0.07 | **0.31**/0.19 |

**Table 2:** Correlation coefficient of lamina number and average vertical resolution at the European mid latitudes stations from the period 1970-2016 (before slash - advective laminae, after slash – gravity wave laminae). Significant correlation coefficient values are in bold.

|  | <1 mPa | 1-4 mPa | >4 mPa |
|---|---|---|---|
| Hohenpeissenberg | 198/203 | 733/1021 | 1895/2057 |
| Payerne | 112/144 | 486/597 | 1874/1803 |
| Uccle | 121/206 | 486/761 | 1832/1775 |
| Legionowo | 104/142 | 535/702 | 1909/1983 |

**Table 3:** Average lamina depth (m) in the selected lamina size intervals at the European midle latitude stations for the vertical resolution below 100m (before slash - advective laminae, after slash – gravity wave laminae).