# Peer review of "Assessing the role of planetary and gravity waves on the vertical structure of ozone over"

_Annales Geophysicae, 2018_

## Referee Comment (RC1) · Anonymous Referee #1 · 4 Jan 2019

The manuscript describes statistics of the lamina appearance in the ozone vertical distribution in dependence of the lamina origin (due to planetary or gravity waves). Thus the subject is well suited to the journal scientific profile. The author uses the methodology elaborated by Teitelbaum et al (1995) to classify the lamina based on the correlation coefficient between vertical profiles of ozone and potential temperature. The reviewer has found interesting and worth publishing results. However, there is a serious problem with selection of the profile data. Thus, the manuscript is not ready for publishing. It may have potential after additional work and resubmission. Table 3 clearly shows that the vertical resolution of the profile should be lower than 100 m for proper identification of the lamina with size less than 1 mPa and less than 500 m

for the lamina size in the range 1-4 mPa. Figure 12 illustrates strong inhomogeneity of the vertical resolution for all the stations. The same is also seen from Table 2. Lindenberg profiles should be excluded from the analysis because of the large and variable vertical resolution. Thus, the analysed data are not homogeneous that may influence the results. A scale of this effect needs to be evaluated in the revised paper or only the latest results with the high resolution of the ozone profiles should be a subject of analysis. It means that the results shown in Fig.6 should be valid for only two stations since 1990 for the lamina size < 1mPa. For laminae in the range 1-4mPa the analyses will be possible for 3 stations since 1970. Thus in present form Fig. 6 is wrong especially for Lindenberg. Minor problems: l.1-2.The title is not proper: Hohenpeissenberg, Payern, and Uccle are located in the western part of Europe. It is better to change the title to "the midlatitudinal Europe". l.112-116. Have you excluded from the analyses evidently wrong profiles with the correction factor far from 1 ( a case for early Legionowo and Lindenberg ozone profiles)? l.158- 185. This section should be rewritten. In fact, Hohenpeissenberg profiles are not proper for analyses of laminae with size <2 mPa as for almost the whole period the vertical resolution is ∼500 m (see Fig.12). The Hohenpeissenberg data are proper for analysis of the laminae with the size > 2 mPa. The author could not state that similar results were derived for other stations, as for Lindenberg (all observations) and Legionowo (early observations before 1990) were not possible to identify correctly lamina with the size <2 mPa. l.190-197. Trend values should appear (% for 10 yr.) with their error estimates to discuss the trend significance. The two-joint lines trend model with the turning point in the mid-1990s needs to applied also for the gravity waves laminae for better comparison with PL laminae. If you calculate the trend based on single line approach for the PL laminae you will probably result with small negative trend as you discussed for the case of the GL lamina trend. l. 215- 220. The discussion is not correct for Payern as this station is located in the valley between the Jura Mountains and Alps.

---

## Referee Comment (RC2) · Anonymous Referee #2 · 31 Jan 2019

Based on the Teitelbaum method, this manuscript studies the characteristics of ozone lamina under the influence of planetary and gravity waves. This article seems to have done a lot of work. Even though I'm not an expert in this area (ozone lamina), there are a few things that make me confused.

Major comments: 1. The formation mechanism of ozone lamina. Tomikawa et al. (2002) reported that the formation of the ozone laminae is closely related to the vertical shear of the subtropical jet. I strongly suggest the authors to discuss in the introduction about the formation mechanisms of the ozone lamina and in which the role of wave activities play.

Tomikawa, Y. , Sato, K. , Kita, K. , Fujiwara, M. , Yamamori, M. , & Sano, T. . (2002). Formation of an ozone lamina due to differential advection revealed by intensive observations. J. Geophys. Res., 107(D10).

2. Lines 101-108, need some references (at least one) or make some explanation: e.g. Why choose 5-15km and 17-22km height area to distinguish. If the identification process is proposed by the author, it appears from the description that the author only uses ozone thin layers at different height region to define whether the ozone laminae is caused by gravity or planetary waves. This makes it very puzzling because gravity waves almost exist anywhere in the earth's atmosphere.

3. The authors have only mentioned the thin layer of ozone caused by gravitational and planetary waves, but I think that some other meso-and small-scale atmospheric processes (such as strong convection, tropopause folding, strong wind shear, stratospheric streamers, etc.) may also responsible for the formation of ozone laminae.

4. Gravity and planetary waves run through the title and the paper, but there is no evidence of their existence in the manuscript (even though the authors indicate that the ozone profile can be used to detect fluctuations)

5. Reading the manuscript, I still didn't understand how gravity and planetary waves affect and lead to ozone laminae. Personally, a detailed case is necessary.

Minor comments: 1. Lines 44-46, as you mentioned, it is the large lamina that has a close correlation with the total ozone content, not the narrow lamina. The actual significance of narrow lamina still not clear throughout the manuscript. 2. Lines 47-48, needs relevant references (at least one), especially about the influence of waves on the laminae. 3. Line 75 from->on ? 4. Line 76 for the approximating-> for approximating 5. Line 125 partitioning of laminae-> partitioning laminae ? 6. Conclusion: as mentioned in the introduction, if the Teitelbaum method is suitable for central Europe? And how well?

---

## Author Comment (AC1) · 13 Mar 2019

Anonymous Referee #2 Based on the Teitelbaum method, this manuscript studies the characteristics of ozone lamina under the influence of planetary and gravity waves. This article seems to have done a lot of work. Even though I'm not an expert in this area (ozone lamina), there are a few things that make me confused. Major comments: 1 The formation mechanism of ozone lamina. Tomikawa et al. (2002) reported that the formation of the ozone laminae is closely related to the vertical shear of the subtropical jet. I strongly suggest the authors to discuss in the introduction about the formation mechanisms of the ozone lamina and in which the role of wave activities play. Tomikawa,

Y. , Sato, K. , Kita, K. , Fujiwara, M. , Yamamori, M. , & Sano, T. . (2002). Formation of an ozone lamina due to differential advection revealed by intensive observations. J. Geophys. Res., 107(D10).

This paper is referred to in Discussion.

2. Lines 101-108, need some references (at least one) or make some explanation: e.g. Why choose 5-15km and 17-22km height area to distinguish. If the identiïfi̧cation process is proposed by the author, it appears from the description that the author only uses ozone thin layers at different height region to defi̧ne whether the ozone laminae is caused by gravity or planetary waves. This makes it very puzzling because gravity waves almost exist anywhere in the earth's atmosphere.

I did the research for all heights from the ground to the 5 km below the highest profile point. The intervals 5-15 km and 17-22 km were chosen because the correlation here is sufficiently high (above 0.7) or low (below 0.3) for detection of gravity and planetary waves. In these intervals the ozone profile are strongly influenced by atmospheric waves. Outside these intervals the profile is not so strongly influenced as in these intervals. The atmospheric waves can occur outside intervals, but they do not influence the ozone vertical profile.

3. The authors have only mentioned the thin layer of ozone caused by gravitational and planetary waves, but I think that some other meso-and small-scale atmospheric processes (such as strong convection, tropopause folding, strong wind shear, stratospheric streamers, etc.) may also responsible for the formation of ozone laminae.

Various mechanisms of lamina formation is described in the Discussion

4. Gravity and planetary waves run through the title and the paper, but there is no evidence of their existence in the manuscript (even though the authors indicate that the ozone profi̧le can be used to detect fluctuations)

This title of paper was recommended by editor.

5. Reading the manuscript, I still didn't understand how gravity and planetary waves affect and lead to ozone laminae. Personally, a detailed case is necessary.

This problem is theoretically solved in Teitelbaum paper.

Minor comments: 1. Lines 44-46, as you mentioned, it is the large lamina that has a close correlation with the total ozone content, not the narrow lamina. The actual signiïficance of narrow lamina still not clear throughout the manuscript.

We were interested in laminae of various sizes because according to theory gravitational waves produce predominantly small size laminae. On the other hand planetary waves are able to form also the large laminae.

2. Lines 47-48, needs relevant references (at least one),especially about the influence of waves on the laminae.

The references were given in the Introduction. 3. Line 75 from->on ?

4. Line 76 for the approximating-> for approximating

This grammar mistakes were corrected

5. Line 125 partitioning of laminae-> partitioning laminae?

We can say partitioning of laminae or lamina partitioning but not partitioning laminae

6. Conclusion: as mentioned in the introduction, if the Teitelbaum method is suitable for central Europe? And how well?

Teitelbaum method was demonstrated for data at the Sodankyla (northern Finland) and this method was able to detect atmospheric waves in the ozone profile. Grant et al. (1998) used the same method for the tropical station\s and this method brought reasonable results. So we suppose this method is suitable also for the stations in Europe, because we obtained results which were expected in the case of well working method,

---

## Author Comment (AC2) · 13 Mar 2019

The manuscript describes statistics of the lamina appearance in the ozone vertical distribution in dependence of the lamina origin (due to planetary or gravity waves). Thus the subject is well suited to the journal scientific profile. The author uses the methodology elaborated by Teitelbaum et al (1995) to classify the lamina based on the correlation coefificient between vertical profiles of ozone and potential temperature. The reviewer has found interesting and worth publishing results. However, there is a serious problem with selection of the profile data. Thus, the manuscript is not ready for publishing. It may have potential after additional work and resubmission. Table 3

clearly shows that the vertical resolution of the profile should be lower than 100 m for proper identification of the lamina with size less than 1 mPa and less than 500 m for the lamina size in the range 1-4 mPa. Figure 12 illustrates strong inhomogeneity of the vertical resolution for all the stations. The same is also seen from Table 2. Lindenberg profiles should be excluded from the analysis because of the large and variable vertical resolution. Thus, the analysed data are not homogeneous that may influence the results. A scale of this effect needs to be evaluated in the revised paper or only the latest results with the high resolution of the ozone profiles should be a subject of analysis. It means that the results shown in Fig.6 should be valid for only two stations since 1990 for the lamina size < 1mPa. For laminae in the range 1-4mPa the analyses will be possible for 3 stations since 1970. Thus in present form Fig. 6 is wrong especially for Lindenberg.

We excluded the station Lindenberg from the paper and we use only the stations Payerne, Uccle and Legionowo in the period 1995-2016 where the vertical resolution of the ozone profile is about 100 m.

Minor problems: l.1-2.The title is not proper: Hohenpeissenberg, Payern, and Uccle are located in the western part of Europe. It is better to change the title to "the midlatitudinal Europe".

The title of the paper was changed

l.112-116. Have you excluded from the analyses evidently wrong profiles with the correction factor far from 1 ( a case for early Legionowo and Lindenberg ozone profiles)?

These profiles were excluded from the analyses.

l.158- 185. This section should be rewritten. In fact, Hohenpeissenberg profiles are not proper for analyses of laminae with size <2 mPa as for almost the whole period the vertical resolution isâĹij500 m (see Fig.12). The Hohenpeissenberg data are proper for

analysis of the laminae with the size > 2 mPa. The author could not state that similar results were derived for other stations, as for Lindenberg (all observations) and Legionowo (early observations before 1990) were not possible to identify correctly lamina with the size <2 mPa.

We use here the station Uccle in the period 1995-2016, so this problem is solved.

l. 190 -197. Trend values should appear (% for 10 yr.) with their error estimates to discuss the trend significance. The two-joint lines trend model with the turning point in the mid1990s needs to apply also for the gravity waves laminae for better comparison with PL laminae. If you calculate the trend based on single line approach for the PL laminae you will probably result with small negative trend as you discussed for the case of the GL lamina trend.

From figure 11 we see principally different trends for PL and GL. So the piecewise regression is suitable only for Pl laminae. This regression is not suitable for GL. In this case it gives insignificant trend before 1995 and insignificant change in 1995. On the other hand the classical regression is erroneous for PL and the most suitable for GL where it gives significant negative trend.

l. 215- 220. The discussion is not correct for Payern as this station is located in the valley between the Jura Mountains and Alps.

This sentence was changed.

Thank you for all your comments. They make my paper better.
* * *

---

## Referee Report (RR1)

The reviewer noticed substantial improvement of the manuscript. However, several minor corrections/additions are necessary to meet quality of the journal.

Minor corrections

1. The remnants of the previous title (…. Central Europe) survived in the new text.

   Replace central Europe to midlatitudinal Europe in the whole manuscript, e.g. l.170, 203, 204, 397.

2. In abstract, PL frequency is 3-5 times larger than GL (l.172) but in the main text the ratio is "about 4-6" (l.391). Which one is correct?  Figure 7 provides that the ratio is around 5 for small lamina but increases for larger lamina, and for large lamina the ratio is about 10. Please provide the values of the ratio for all lamina classes. It is not enough to say that " with the increasing lamina size the share of GL decreases and the share of PL increases" (l.174-175).

3. L.175-176.  Consider changes in the statement: "The vertical profile of lamina occurrence is different for small planetary wave and gravity wave laminae". Use abbreviations PL and  GL. The difference concerns all (large/small) laminae. Be more precise and define what the mentioned difference is and how it depends on lamina size?

4. l. 238 "the average ozone profile (potential temperature)" Please define meaning of the "average" i.e. the mean for month, season, year etc.

5. l.245. Start new line beginning with  "In each point……"

6. l.293-294 Consider rewriting: " If these correlations are significant the resolution influences the lamina number and vice versa". Delete "and vice versa" In fact,  number of lamina does not affect technical issue of the ozone-sonde resolution.

7. l.295. The results are shown in Table 2 not in Table 1.

8. L. 646 and l. 651. Tab.1 and Tab.2 . Term advective lamina is used. It should be PL.

9. l.301. It should be  Tab.3  instead Tab.2.

10. l.308-310. Delete these lines and start with "The vertical resolution of sonde **measurements** must be….."  Please add  "measurements" after "sonde".

11. l.311. It should be Table 3 instead Tab.2.

12. l.325-326. It is better to say that "Annual variation with the maximum in winter/spring and summer/autumn minimum is clearly seen for PL but this pattern is very weak in case of GL". Section 3.3 is very short, so add some comments that the maximum is about 4 times higher than the minimum in case of PL. Please also discuss the yearly mean values in both cases.

13. L.323 and l. 332 – It should be different numbers not both 3.3. Further changes are needed for all subsection numbers in section 3.

14. L.628. Here Figure number should be 11

15. Fig8-Fig.11 . For better comparison the range of X axis should be the same for all figures, e.g. [0,30].

16. Section 3.4. The results are shown for Uccle only.  The reviewer would like to see results for Payerne as this station is located in the valley between the Jura Mountains and the Alps and it seems that GL profile will be different in the troposphere.

17. There is a serious problem with section 3.5 – Trend of large laminae
    Why only the results from Hohenpeissenberg have been analysed?
    There is possibility of trend analyses of all types of laminae for Uccle and Payerne starting in 1990 and for Legionowo starting in 1995. Simple linear regression should be used in this case. For comparison purposes it is better to focus on trend for the period 1995-2016.

Piecewise approach is valid for longer data- Hohenpeissenberg 1970-2016. Here you present trend results for only one station. It is not mentioned in the abstract, conclusions, and in section 3.5. You cannot envisage that similar trend patter appears for other stations. Moreover, the trend analysis is not mentioned in section 4 (Discussion). Your trend analysis should contain more stations even with shorter data if your interest is the long-term lamina variability over the midlatitudinal Europe. My recommendation is to delete this section and omit discussion concerning the trends (l.176-177) and l. 462.  More comprehensive analysis of long-term variability of laminae over the midlatitudinal Europe is a good subject for next your paper.

18. L.425. Be more precise. What is the meaning of small GL maximum? Another maximum or the maximum for small GL(<2hPa)?

19. L.425-427. " In summer the occurrence …… It is not clear what kind of laminea you describe here, GL or PL. For me it seems that is valid only for PL.
It is better to limit the discussion to apparent maxima as you have a plenty of secondary extremes.

20. L.424-425 "occurrence maximum is observed in the tropopause". It is better to say near tropopause as you have no info about the tropopause height.

21. L403-405. At this point the reviewer would like to see a discussion of gravity waves over Payerne (a site between the Jura Mountains and the Alps). See also problem no.16.

---

## Author Response (AR3)

The manuscript describes statistics of the lamina appearance in the ozone vertical distribution in dependence of the lamina origin (due to planetary or gravity waves). Thus the subject is well suited to the journal scientific profile. The author uses the methodology elaborated by Teitelbaum et al (1995) to classify the lamina based on the correlation coefficient between vertical profiles of ozone and potential temperature. The reviewer has found interesting and worth publishing results. However, there is a serious problem with selection of the profile data. Thus, the manuscript is not ready for publishing. It may have potential after additional work and resubmission. Table 3 clearly shows that the vertical resolution of the profile should be lower than 100 m for proper identification of the lamina with size less than 1 mPa and less than 500 m for the lamina size in the range 1-4 mPa. Figure 12 illustrates strong inhomogeneity of the vertical resolution for all the stations. The same is also seen from Table 2. Lindenberg profiles should be excluded from the analysis because of the large and variable vertical resolution. Thus, the analysed data are not homogeneous that may influence the results. A scale of this effect needs to be evaluated in the revised paper or only the latest results with the high resolution of the ozone profiles should be a subject of analysis. It means that the results shown in Fig.6 should be valid for only two stations since 1990 for the lamina size < 1mPa. For laminae in the range 1-4mPa the analyses will be possible for 3 stations since 1970. Thus in present form Fig. 6 is wrong especially for Lindenberg.

We excluded the station Lindenberg from the paper and we use only the stations Payerne, Uccle and Legionowo in the period 1995-2016 where the vertical resolution of the ozone profile is about 100 m.

Minor problems: l.1-2.The title is not proper: Hohenpeissenberg, Payern, and Uccle are located in the western part of Europe. It is better to change the title to "the midlatitudinal Europe".

The title of the paper was changed

l.112-116. Have you excluded from the analyses evidently wrong profiles with the correction factor far from 1 ( a case for early Legionowo and Lindenberg ozone profiles)?

These profiles were excluded from the analyses.

l.158- 185. This section should be rewritten. In fact, Hohenpeissenberg profiles are not proper for analyses of laminae with size <2 mPa as for almost the whole period the vertical resolution is~500 m (see Fig.12). The Hohenpeissenberg data are proper for analysis of the laminae with the size > 2 mPa. The author could not state that similar results were derived for other stations, as for Lindenberg (all observations) and Legionowo (early observations before 1990) were not possible to identify correctly lamina with the size <2 mPa.

We use here the station Uccle in the period 1995-2016, so this problem is solved.

l. 190 -197. Trend values should appear (% for 10 yr.) with their error estimates to discuss the trend significance. The two-joint lines trend model with the turning point in the mid1990s needs to apply also for the gravity waves laminae for better comparison with PL laminae. If you calculate the trend based on single line approach for the PL laminae you will probably result with small negative trend as you discussed for the case of the GL lamina trend.

From figure 11 we see principally different trends for PL and GL. So the piecewise regression is suitable only for Pl laminae. This regression is not suitable for GL. In this case it gives insignificant trend before 1995 and insignificant change in 1995. On the other hand the classical regression is erroneous for PL and the most suitable for GL where it gives significant negative trend.

l. 215- 220. The discussion is not correct for Payern as this station is located in the valley between the Jura Mountains and Alps.

This sentence was changed.

Thank you for all your comments. They make my paper better.
Based on the Teitelbaum method, this manuscript studies the characteristics of ozone lamina under the influence of planetary and gravity waves. This article seems to have done a lot of work. Even though I'm not an expert in this area (ozone lamina), there are a few things that make me confused.

Major comments: 1 The formation mechanism of ozone lamina. Tomikawa et al. (2002) reported that the formation of the ozone laminae is closely related to the vertical shear of the subtropical jet. I strongly suggest the authors to discuss in the introduction about the formation mechanisms of the ozone lamina and in which the role of wave activities play. Tomikawa, Y. , Sato, K. , Kita, K. , Fujiwara, M. , Yamamori, M. , & Sano, T. . (2002). Formation of an ozone lamina due to differential advection revealed by intensive observations. J. Geophys. Res., 107(D10).

This paper is referred to in Discussion.

2. Lines 101-108, need some references (at least one) or make some explanation: e.g. Why choose 5-15km and 17-22km height area to distinguish. If the identification process is proposed by the author, it appears from the description that the author only uses ozone thin layers at different height region to define whether the ozone laminae is caused by gravity or planetary waves. This makes it very puzzling because gravity waves almost exist anywhere in the earth's atmosphere.

I did the research for all heights from the ground to the 5 km below the highest profile point. The intervals 5-15 km and 17-22 km were chosen because the correlation here is sufficiently high (above 0.7) or low (below 0.3) for detection of gravity and planetary waves. In these intervals the ozone profile are **strongly** influenced by atmospheric waves. Outside these intervals   the profile is not so strongly influenced as in these intervals. The atmospheric waves can occur outside intervals, but they do not influence the ozone vertical profile.

3. The authors have only mentioned the thin layer of ozone caused by gravitational and planetary waves, but I think that some other meso-and small-scale atmospheric processes (such as strong convection, tropopause folding, strong wind shear, stratospheric streamers, etc.) may also responsible for the formation of ozone laminae.

Various mechanisms of lamina formation is described in the Discussion

4. Gravity and planetary waves run through the title and the paper, but there is no evidence of their existence in the manuscript (even though the authors indicate that the ozone profile can be used to detect fluctuations)

This title of paper was recommended by editor.

5. Reading the manuscript, I still didn't understand how gravity and planetary waves affect and lead to ozone laminae. Personally, a detailed case is necessary.

This problem is theoretically solved in Teitelbaum paper.

Minor comments: 1. Lines 44-46, as you mentioned, it is the large lamina that has a close correlation with the total ozone content, not the narrow lamina. The actual significance of narrow lamina still not clear throughout the manuscript.

We were interested in laminae of various sizes because according to theory gravitational waves produce predominantly small size laminae. On the other hand planetary waves are able to form also the large laminae.

2. Lines 47-48, needs relevant references (at least one),especially about the influence of waves on the laminae.

The references were given in the Introduction.

3. Line 75 from->on ?

4. Line 76 for the approximating-> for approximating

This grammar mistakes were corrected

5. Line 125 partitioning of laminae-> partitioning laminae?

We can say partitioning of laminae or lamina partitioning but not partitioning laminae

6. Conclusion: as mentioned in the introduction, if the Teitelbaum method is suitable for central Europe?  And how well?

Teitelbaum method was demonstrated for data at the Sodankyla (northern Finland) and this method was able to detect atmospheric waves in the ozone profile. Grant et al. (1998) used the same method for the tropical station\s  and this method brought  reasonable results. So we suppose this method is suitable also for the stations in Europe, because we obtained results which were expected in the case of well working method.

The reviewer noticed substantial improvement of the manuscript. However, several minor corrections/additions are necessary to meet quality of the journal.

Minor corrections

1. The remnants of the previous title (…. Central Europe) survived in the new text.

Replace central Europe to midlatitudinal Europe in the whole manuscript, e.g. l.170, 203, 204, 397.

All remnants were replaced.

2. In abstract, PL frequency is 3-5 times larger than GL (l.172) but in the main text the ratio is "about 4-6" (l.391). Which one is correct? Figure 7 provides that the ratio is around 5 for small lamina but increases for larger lamina, and for large lamina the ratio is about 10. Please provide the values of the ratio for all lamina classes. It is not enough to say that " with the increasing lamina size the share of GL decreases and the share of PL increases" (l.174-175).

I did table which displayed the ratio of PL and GL in each month and station used in paper and the results are discussed in paper.

3. L.175-176. Consider changes in the statement: "The vertical profile of lamina occurrence is different for small planetary wave and gravity wave laminae". Use abbreviations PL and GL. The difference concerns all (large/small) laminae. Be more precise and define what the mentioned difference is and how it depends on lamina size?

We did it

4. l. 238 "the average ozone profile (potential temperature)" Please define meaning of the "average" i.e. the mean for month, season, year etc.

5. l.245. Start new line beginning with "In each point……"

6. l.293-294 Consider rewriting: " If these correlations are significant the resolution influences the lamina number and vice versa". Delete "and vice versa" In fact, number of lamina does not affect technical issue of the ozone-sonde resolution.

7. l.295. The results are shown in Table 2 not in Table 1.

8. L. 646 and l. 651. Tab.1 and Tab.2 . Term advective lamina is used. It should be PL.

9. l.301. It should be Tab.3 instead Tab.2.

10. l.308-310. Delete these lines and start with "The vertical resolution of sonde measurements must be….." Please add "measurements" after "sonde".

11. l.311. It should be Table 3 instead Tab.2.

Points 4-11 were changed according to reviewer suggestions.

12. l.325-326. It is better to say that "Annual variation with the maximum in winter/spring and summer/autumn minimum is clearly seen for PL but this pattern is very weak in case of GL". Section 3.3 is very short, so add some comments that the maximum is about 4 times higher than the minimum in case of PL. Please also discuss the yearly mean values in both cases.

We did it

13. L.323 and l. 332 – It should be different numbers not both 3.3. Further changes are needed for all subsection numbers in section 3.

14. L.628. Here Figure number should be 11

15. Fig8-Fig.11 . For better comparison the range of X axis should be the same for all figures, e.g. [0,30].

Points 13-15 were changed according to reviewer suggestions.

16. Section 3.4. The results are shown for Uccle only.  The reviewer would like to see results for Payerne as this station is located in the valley between the Jura Mountains and the Alps and it seems that GL profile will be different in the troposphere.

We show the results for the station Payerne, but these results were not different from the other stations.

17.  There is a serious problem with section 3.5 – Trend of large laminae Why only the results from Hohenpeissenberg have been analysed? There is possibility of trend analyses of all types of laminae for Uccle and Payerne starting in 1990 and for Legionowo starting in 1995. Simple linear regression should be used in this case. For comparison purposes it is better to focus on trend for the period 1995-2016.

Piecewise approach is valid for longer data- Hohenpeissenberg 1970-2016. Here you present trend results for only one station. It is not mentioned in the abstract, conclusions, and in section 3.5. You cannot envisage that similar trend patter appears for other stations. Moreover, the trend analysis is not mentioned in section 4 (Discussion). Your trend analysis should contain more stations even with shorter data if your interest is the long-term lamina variability over the midlatitudinal Europe. My recommendation is to delete this section and omit discussion concerning the trends (l.176-177) and l. 462.  More comprehensive analysis of long-term variability of laminae over the midlatitudinal Europe is a good subject for next your paper.

Trends of laminae was deleted from the paper.

 18. L.425. Be more precise. What is the meaning of small GL maximum? Another maximum or the maximum for small GL(<2hPa)?

19. L.425-427. " In summer the occurrence …… It is not clear what kind of laminea you describe here, GL or PL. For me it seems that is valid only for PL.  It is better to limit the discussion to apparent maxima as you have a plenty of secondary extremes.

20. L.424-425 "occurrence maximum is observed in the tropopause". It is better to say near tropopause as you have no info about the tropopause height.

The paragraph concerning the vertical profile of lamina occurrence was rewritten.

21. L403-405. At this point the reviewer would like to see a discussion of gravity waves over Payerne (a site between the Jura Mountains and the Alps). See also problem no.16.

We discuss this topic in the paper.

[revised manuscript text omitted]